# Generalizing Robustness from $\ell_p$ to Unforeseen Attack via Calibrated Adversarial Sampling

## Abstract

Deep Neural Networks (DNNs) are known to be vulnerable to various adversarial perturbations. To address the safety concerns arising from these vulnerabilities, adversarial training (AT) has emerged as one of the most effective paradigms for enhancing the robustness of DNNs. However, existing AT frameworks primarily focus on a single or a limited set of attack types, leaving DNNs still exposed to newly considered attack types that have not been addressed during training. In this paper, we explore a new robust generalization paradigm that fine-tunes robust DNNs to cope with unforeseen attacks. To this end, we propose Calibrated Adversarial Sampling (CAS), a method that dynamically adjusts sampling probabilities during fine-tuning to balance robustness across various adversarial attacks. CAS operates in three key phases: sample-wise robustness testing, warm-up fine-tuning, and dynamic fine-tuning. Experiments on benchmark datasets show that CAS achieves superior overall robustness, maintains clean accuracy, and effectively balances robustness across different types of attacks, providing a new paradigm for robust generalization of DNNs.

## 1 Introduction

Deep Neural Networks (DNNs) are known to be vulnerable against adversarial attacks [1], where attackers can add imperceptible [29, 9] or semantic [11, 12] perturbations to craft adversarial examples that lead the target DNN to make incorrect predictions. So far, the existence of adversarial examples has raised significant concerns about DNNs [19, 17, 20], compromising their trustworthiness in their deployments.

To address these concerns, numerous defense tactics have been proposed, such as adversarial training (AT) [23, 36, 33], robustness repair [28, 21, 4], and adversarial noise purification [35, 25, 3]. Despite their success in different deployment stages of DNNs, most of the existing defenses only focus on a particular robustness metric. However, in this context, the metrics of DNN robustness can be diverse. Generally, adversarial perturbations can be categorized into (i) $\ell_p$-**norm perturbations**, and (ii) **semantic perturbations**. The $\ell_p$-norm perturbation $\delta$ is commonly optimized through the classification loss (*e.g.*, cross-entropy) and constrained by an $\ell_p$ ball $\|\delta\|_p \le \epsilon$, where popular $p \in \{1, 2, \infty\}$. They are more imperceptible due to the $\ell_p$-norm constraint, yet are difficult to directly inject into real-world vision models. By contrast, semantic perturbation is crafted by adding preset transformation rules, *e.g.*, snow, geometric transformation, etc.. Examples of these two kinds of perturbations are illustrated in Figure 1.

The current adversarially robust generalization literature primarily focuses on the $\ell_p$-norm robustness of DNNs. For example, various AT techniques are designed toward a single worst-case $\ell_\infty$ or $\ell_2$-norm adversarial robustness [23, 27, 8, 31]. Though a few preliminary works focus on multiple robustness metrics, they are either limited to multiple $\ell_p$-norm robustness [6] or require training the model from

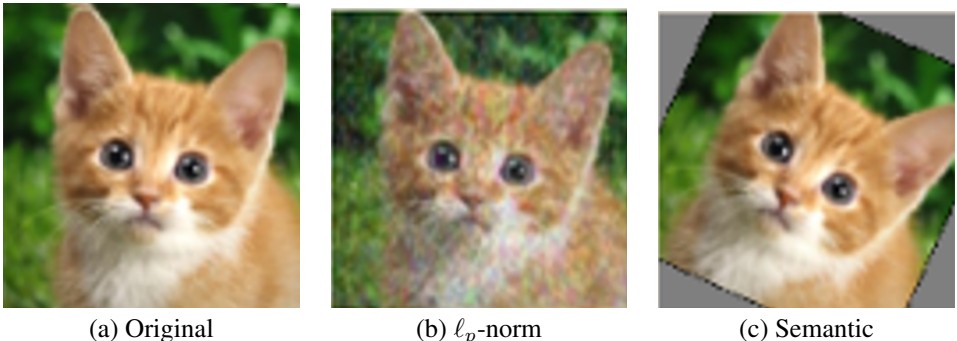

| (a) Original | (b) $\ell_p$-norm | (c) Semantic |

Figure 1: Illustration of adversarial perturbations by different attack types from [12].

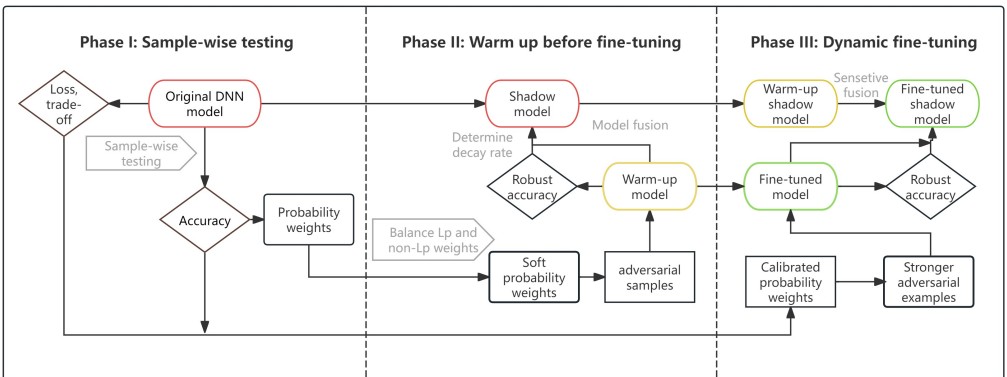

Figure 2: An overview of CAS. CAS operates in three sequential phases. It first evaluates model performance: accuracy and robustness across various adversarial attacks. CAS then warms up multi-robustness fine-tuning by sampling attacks with smoothed probability weights. Finally, it dynamically updates the sampling probabilities via synthesizing previously measured metrics.

scratch [24]. In this paper, we explore a new robust generalization paradigm through fine-tuning $\ell_p$-norm robust DNNs. Specifically, we aim to fine-tune robust DNNs against unforeseen attacks (*i.e.*, adversarial attacks that are not employed during the initial training phase).

The motivation behind this paradigm is illustrated as follows. First, existing AT techniques designed for multiple robustness metrics primarily focus on the $\ell_p$-norm perturbations. However, novel and unforeseen attacks, particularly those involving semantic perturbations, remain largely unexplored. Additionally, while these techniques can be directly adapted to defend both $\ell_p$ and semantic perturbations, they necessitate training from scratch and struggle to generalize to newly identified threats continuously. Furthermore, since numerous pre-trained $\ell_p$-norm robust models are available online for commonly used datasets like RobustBench [18], it is feasible to leverage and fine-tune these models against new attacks rather than training from scratch, as conducting AT on DNNs requires expensive computational cost.

To this end, we introduce a technique called Calibrated Adversarial Sampling (CAS), which aims to balance the trade-offs in sampling among various adversarial attacks during fine-tuning robust DNNs. Specifically, CAS employs dynamic probability weights during fine-tuning to holistically optimize resource allocation by integrating trade-off considerations regarding different adversarial perturbations. This procedure includes three phases: (i) sample-wise robustness testing to obtain the calibrated probability weights for adversarial attacks sampling, (ii) warm-up fine-tuning to facilitate the model with basic robustness of newly introduced attacks, and (iii) dynamic fine-tuning to balance better trade-offs between different robustness metrics. Additionally, given the notable success of the Exponential Moving Average (EMA) mechanism in improving adversarial robustness [30, 33], we design an EMA variant tailored for multiple robustness trade-offs to enhance overall accuracy further.

To validate the effectiveness of CAS, we conduct extensive experiments on pre-trained robust models trained on popular benchmark datasets that are available from RobustBench [18]. For

adversarial attacks, we consider three common $\ell_p$ attacks and 18 semantic attacks (*e.g.*, snow, fog, blur). Experimental results show CAS achieves superior overall robustness and maintains the clean accuracy, and also balances robustness against different types of attacks, contributing to a novel and practical paradigm of DNN robust generalization. Overall, our contribution in this paper can be summarized as follows:

- We explore a new DNN robust generalization paradigm that fine-tunes $\ell_p$-robust DNNs for unforeseen semantic adversarial attacks and propose Calibrated Adversarial Sampling (CAS) to address this issue.

- CAS extends and innovates upon the dynamic probability weighting methodology from $\ell_p$-robustness fine-tuning baselines, achieving a dual breakthrough in both methodological depth and breadth by comprehensively addressing both $\ell_p$ and semantic adversarial attacks.

- Extensive experiments demonstrate that CAS achieves superior effectiveness and efficiency towards this goal compared to conventional methods, offering a novel technique in real-world robust deployments of DNNs.

## 2 Related Work

### 2.1 Adversarial Examples

Adversarial examples are firstly discovered as deceptive samples crafted by applying subtle (often imperceptible) perturbations to clean inputs [29], which can mislead DNNs into making erroneous predictions. In practice, such perturbations are typically constrained within specific norm balls (*e.g.*, $\ell_p$-norm constraints). To deal with this threat, *adversarial training* has emerged as the primary defense paradigm [34, 2, 27], which enhances model robustness by explicitly injecting adversarial examples during training.

Notably, beyond $\ell_p$-bounded "pixel-level" perturbations, there exists a challenging class of *semantic adversarial examples*. These leverage natural, semantically meaningful transformations (*e.g.*, rotations, translations, lighting changes, fog, and blur) to deceive models, operating in larger perturbation spaces that better reflect real-world variations [11]. Correspondingly, *semantic adversarial training* has emerged as a critical research direction to enhance robustness against such semantically valid perturbations [12].

### 2.2 Multi-Robustness of DNNs

While single-type adversarial training targets specific attack models, comprehensive adversarial training for *multi-robustness* is more effective in evaluating and enhancing a model's defenses against a broader range of attacks, providing stronger and more practical protection in real-world scenarios [7]. Moreover, fine-tuning techniques can further improve multi-robustness by building on existing models, making them highly applicable in practice.

**Adversarial Training**. Adversarial training (AT) methods share a common limitation: training a model from scratch requires substantial computational resources and disregards progress made through prior model development. Stochastic adversarial training (SAT) [22] injects random noise during adversarial example generation to enhance robustness against $\ell_1$, $\ell_2$, and $\ell_\infty$ on AutoAttack benchmarks. This approach generalizes well to unforeseen attacks. Similarly, Multi-perturbation adversarial training (MPAT) [24] achieves strong multi-norm robustness by jointly optimizing against $\ell_1$, $\ell_2$, and $\ell_\infty$ perturbations using a multi-steepest descent method.

**Fine-Tuning**. Fine-tuning methods offer a more computationally efficient alternative to full retraining for enhancing robustness across multiple threat models. E-AT [6], a fast fine-tuning method for robust classifiers, significantly boosts robustness across $\ell_p$ norms with minimal training cost. It achieves this by: (1) utilizing extreme norms ($\ell_1$, $\ell_\infty$) to implicitly cover intermediate ($\ell_p$, $p > 1$) perturbations; (2) and applying dynamic probability weighting based on average error rates. RAMP [14] introduces a regularization-based fine-tuning framework that uses a logit pairing loss function to account for the effects of different $\ell_p$ attacks. It achieves a union accuracy of up to 53.3% on CIFAR-10 and 29.1% on ImageNet.

However, both E-AT and RAMP fine-tuning focus solely on $\ell_p$-norm adversarial attacks. In contrast, our proposed CAS framework integrates defenses against both $\ell_p$ perturbations and semantic adversarial attacks, providing a more comprehensive and unified approach to fine-tuning for multi-robustness.

## 3 Preliminaries

### 3.1 Unforeseen Adversarial Attacks

While significant breakthroughs have currently been achieved in research on $\ell_p$ robustness, such as AutoAttack [5], systematic investigations into semantic adversarial attacks remain comparably under-explored [5, 20]. Moreover, current robustness evaluations are often confined to single attack types, lacking a holistic framework that jointly considers diverse threat models. This narrow focus poses significant limitations for real-world AI deployments (*e.g.*, autonomous driving, image processing systems), where robustness against physically plausible and naturalistic corruptions is critical for safety-critical applications [11, 32].

To enhance the versatility and comprehensiveness of robust fine-tuning methods in real-world applications, this work investigates *multi-robustness* under a broad and diverse set of adversarial attacks. From the framework proposed by [15], we select 17 representative semantic attacks for evaluation, including *Wood, Elastic, Pixel, Snow, Gabor, JPEG, Glitch, Kaleidoscope, Blur, Edge, Fog, Texture, Prison, Whirlpool, Polkadot, Klotski, and Hsv*. These cover a wide range of both environmental and digital perturbations that commonly arise in real-world settings. In addition, we incorporate the PerceptivePGD (PPGD) attack [37], a human-perception-guided method that explicitly models semantically meaningful visual changes, along with three standard $\ell_p$-norm attacks: $\ell_\infty$, $\ell_2$, and $\ell_1$. Together, this suite comprises 21 distinct adversarial attacks, enabling a comprehensive evaluation of model robustness against both semantic and norm-constrained perturbations.

### 3.2 Mutually Exclusive Perturbations

Since our research comprehensively considers 21 adversarial attacks to transcend the limitations of prior studies that were predominantly confined to $\ell_p$ perturbations, integrating diverse adversarial attacks into a unified adversarial training framework introduces significant challenges. In particular, the Mutually Exclusive Perturbations (MEPs) [15] theory represents a fundamental limitation. MEPs occur when the constraint sets of two perturbation types are inherently incompatible, such that improving robustness against one attack inevitably degrades robustness against the other under fixed optimization conditions. Classic examples of this contradiction include the $\ell_p$ attack and the rotation-translation transformation as discussed in [15].

While the concept of MEPs *qualitatively* captures the inherent conflicts in achieving multi-attack robustness, it currently lacks a rigorous definition and quantitative characterization. In the following, we aim to formalize and extend this concept through *quantitative* analysis.

### 3.3 Quantifying Multi-Robustness Trade-Off

To systematically quantify these phenomena, we conduct a large-scale empirical study based on the online pre-trained $\ell_\infty$-robust model (called pretr_Linf) [6] on the CIFAR-10 dataset [16], since pretr_Linf exhibits inherent robustness against many adversarial attacks compared to a completely non-robust pre-trained model. In this experiment, we choose 11 perturbation types – $\ell_\infty$, $\ell_2$, $\ell_1$, and 8 semantic attacks. For each attack type, we perform individual adversarial fine-tuning for 10 epochs and measure the robust accuracy against all 11 attacks before/after fine-tuning. As illustrated in Figure 3, we observe several notable patterns:

**Semantic-$\ell_p$ Conflicts**. As shown in the figure, semantic attacks often degrade $\ell_p$ robustness because most natural corruptions are incompatible with $\ell_p$ perturbation constraints.

**Transfer Asymmetry**. The tradeoff matrix shows no symmetric pattern with respect to the diagonal. For instance, $\ell_p$ adversarial training enhances robustness against most semantic attacks, while adversarial training using most semantic attacks tends to impair $\ell_p$ robustness.

**Robustness Interference**. When summing all values in the tradeoff matrix, it yields a notably negative total of $-3.123$, indicating that sequential training against individual attack types tends to

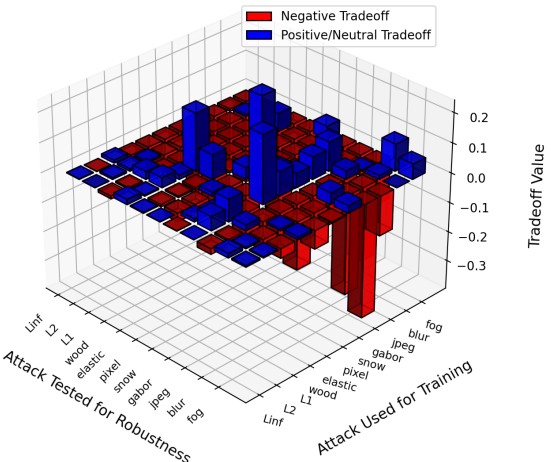

Figure 3: Trade-off matrix visualization. In the figure, each bar represents the change in robust accuracy for its corresponding attack type after AT against designated types.

decrease overall robustness. This mutual interference of robustness presents a fundamental challenge for improving multi-robustness. Interestingly, probability-weighted training methods such as SAT demonstrate improved average robust accuracy by random sampling adversarial attacks during training. This highlights the critical role of training sample order in multi-robustness optimization.

These findings underscore that the mutual interference dilemma for MEPs cannot be freely solved. Based on this matrix, we propose the Calibrated Adversarial Sampling (CAS) method, which leverages empirical data from a trade-off matrix to quantify the "externality" of each adversarial attack—defined as its cross-impact on other robustness types—thereby enabling systematic optimization of robustness allocation across diverse threat categories.

# 4 Methodology

This section details our CAS framework. We begin with a brief overview, followed by a detailed description of its three key phases.

## 4.1 Framework Overview

As illustrated in Figure 2, the CAS framework operates in three phases. First, it evaluates the accuracy, loss, and trade-off for each batch of data under various adversarial attacks to provide a prior for the subsequent probability calibration. Next, CAS selects adversarial attacks randomly with calibrated probability weights to warm up the Multi-Robustness fine-tuning. Finally, CAS synthesizes the previously evaluated accuracy, loss, and trade-off to dynamically update the probability weights.

## 4.2 Phase I: Sample-Wise Testing

Since quantitative balancing across different attack types is required for CAS, (e.g., using the aforementioned trade-off matrix), the weighting parameters must be derived from testing data. This initial phase establishes a granular vulnerability assessment framework, designed to quantify the DNN's robustness at the individual sample level. Unlike conventional model-wide evaluations, our sample-wise approach enables sample-wise precise identification of vulnerabilities, forming the foundation for targeted fine-tuning in subsequent phases.

For each adversarial attack type $A_j$, we compute per-batch metrics including mean robust accuracy $\mathrm{acc}[j]$ and cross-entropy loss $\mathrm{loss}[i][j]$.

$$x'_{ij} = A_j(x_i, \epsilon_j) \quad \text{subject to} \quad D_j(x'_{ij} - x_i) \leq \epsilon_j \tag{1}$$

$$\text{acc}[j] = \frac{1}{n} \sum_{i=1}^{n} \mathbf{I}\big(N(x'_{ij}) = y_{ij}\big) \tag{2}$$

$$\text{loss}[i][j] = \text{CrossEntropy}(N(x'_{ij}), y_{ij}) \tag{3}$$

where $D_j(x'_{ij} - x_i)$ denotes the distance between $x'_{ij}$ and $x_i$ under the perturbation constraints of $A_j$. Concurrently, we record robust accuracy across all attack types before training (acc_b$_j$) and after training (acc_a$_j$) against $A_j$ attack. The pairwise trade-off matrix is then derived as:

$$\text{TR}[j][k] = \text{acc\_a}_j[k] - \text{acc\_b}_j[k] \tag{4}$$

However, substantial computational overhead emerges when handling numerous adversarial attack types, with the tradeoff matrix summation $\sum_{j,k} TR[j][k]$ typically being slightly negative. This indicates that sample-wise testing is computationally intensive and suboptimal for model performance. To mitigate computational costs, Phase I executes just one epoch where attack $A_j$ is exclusively evaluated on batches satisfying $i \mod \text{len}(\mathcal{A}) = j$ ($\mathcal{A}$ being the attack set). This partial evaluation creates incomplete loss data $loss[i][j]$, so we initialize all $loss[i][j] = 2.0$ (rather than 0.0) and excludes loss metrics from probability weight calculations during warm-up phase to minimize the impact of missing values.

### 4.3 Phase II: Warm-Up Fine-Tuning with Accuracy-Driven Optimization

This phase initiates model adaptation using accuracy-based probabilistic weighting. The selection probability for attack $A_j$ follows:

$$p[j] = \exp\left(1 - \text{acc}[j]\right) \tag{5}$$

with categorical reweighting for $\ell_p$ attacks:

$$p[j] = p[j] \times \frac{m - m_p}{m}, \ \forall j \ s.t. \ A_j \text{ is an } \ell_p \text{ attack} \tag{6}$$

This scaling preserves relative selection frequency for $\ell_p$ attack families despite varying $m_p/m$ ratios, preventing optimization bias toward more numerous attack types. The effectiveness of $\ell_p$/semantic reweighting is validated through ablation studies in the next section.

Meanwhile, since research has found that EMA (Exponential Moving Average) substantially enhances adversarial robustness [13, 30, 33], we explore adapting the EMA techniques specialized for the multiple robustness setting. The vanilla EMA update during training can be formulated as:

$$\bar{\theta} = \alpha\bar{\theta} + (1 - \alpha)\theta \tag{7}$$

where $\alpha$ is the decay rate, $\theta$ is the parameter of the model at the current epoch, and $\bar{\theta}$ is the parameter of the EMA model.

Conventionally, $\alpha$ is a fixed hyper-parameter. While in this setting, we propose a Multi-robustness-oriented Dynamic EMA (MDE) method, where the decay rate is adapted to trade-off magnitude:

$$\alpha = \min\left(\tau, 1 - \frac{d_2}{m} \sum_{k=1}^{m} (\text{acc\_a}[k] - \text{acc\_b}[k])\right) \tag{8}$$

where $d_2$ is a hyper-parameter and $\tau$ denotes a fixed threshold. Selecting a threshold marginally below 1 is to prevent the model from converging to local optima during MDE.

Moreover, the adversarial loss matrix $loss[i][j]$ and robustness accuracy vector $acc[j]$ are dynamically updated after per-batch training.

### 4.4 Phase III: Dynamic Fine-Tuning for Balancing Robustness Trade-Offs

After the warm-up phase, we proceed to conduct finer-grained and better-calibrated targeted fine-tuning.

The core optimization phase employs a multi-criteria weighting scheme:

$$p[j] = \exp\left(w_1(1 - acc[j]) + w_2 loss[i][j] + w_3 V_{TR}\right) \tag{9}$$

where $w_1, w_2, w_3$ are hyper-parameters and $V_{TR}$ denotes the trade-off based vulnerability matrix (*i.e.* $sum(TR[j])$). This phase features a significantly steeper probability weight distribution than during warm-up.

In this phase, MDE maintenance continues with decay rate scaled for diminishing returns:

$$\alpha = \min \left( \tau, 1 - \frac{d_3}{m} \sum_{k=1}^{m} (\text{acc\_a}[k] - \text{acc\_b}[k]) \right) \tag{10}$$

where $d_3$ is a hyper-parameter much larger than $d_2$. Because during the fine-tuning phase, robustness improvements are generally smaller than those achieved in the warm-up phase.

### 4.5 Summary and Discussion

The complete algorithm of our method can be summarized in appendix. Collectively, the CAS methodology demonstrates qualitative superiority by holistically considering and balancing $\ell_p$ and semantic attacks, outperforming both E-AT and RAMP in semantic attack robustness. Its dynamic probability weighting mechanism proves more effective than static-weight SAT when facing non-uniform perturbation intensity distributions across different attacks. Furthermore, CAS's integrated consideration of accuracy, loss, and trade-off in probability weighting fundamentally distinguishes it from E-AT-all extensions that solely rely on accuracy metrics. The Multi-robustness-oriented Dynamic EMA (MDE) variant further enhances CAS's efficacy beyond conventional EMA integrations. We will quantitatively validate these advantages in the subsequent experimental section.

## 5 Experiment

In this section, we demonstrate the effectiveness of our proposed CAS framework to improve overall robustness and address the Multi-Robust Trade-off.

### 5.1 Experimental Setup

We conduct our experiments on the benchmark dataset CIFAR-10 and CIFAR-100 [16] using the pre-trained PreActResNet-18 [10] models (pretr_Linf, pretr_L2, and pretr_L1) provided by [6]. We consider the 21 different adversarial attacks during fine-tuning presented in the Preliminary section.

**Baselines**. We select E-AT [6], RAMP [14], and SAT [22] as our baselines, which are introduced in the Related Work section. While E-AT solely considers $\ell_p$ robustness, we also consider its extension form, which is an expanded baseline incorporating 21 adversarial attacks, called E-AT-all. Additionally, since our CAS method is a variant of the weight average method with EMA, we add the EMA method to our baselines to ensure a fair comparison. In particular, since RAMP inherently incorporates the innovation of model fusion, we refrain from augmenting it with additional attack types or EMA methodology.

**Training Settings**. Following the common practice of AT [26, 30, 33], we fine-tune a pre-trained PRN-18 model using SGD with momentum 0.9, weight decay $5 \times 10^{-4}$, and initial learning rate 0.1 for 20 epochs. All $\ell_p$ attacks are conducted by the default perturbation margin $\epsilon_\infty = \frac{8}{255}, \epsilon_2 = 0.5, \epsilon_1 = 12$. For semantic attacks, we We employ an iteratively computed calibrated margin to ensure that most adversarial perturbations maintain accuracy between 20% and 60% on the pre-trained $\ell_\infty$-robust model on CIFAR-10:

$$\epsilon_k = (\lambda_k + acc_{adv}[k]) * \epsilon \tag{11}$$

Where $\epsilon_k$ is the original perturbation margin, $\lambda_k$ is a hyper-parameter to ensure that the robust accuracy $acc_{adv}[k]$ is between 20% and 60%.

**Evaluation Metrics**. We evaluate the clean and robust accuracy in the average case against different attacks. The $\ell_p$ robustness is evaluated by AutoAttack [5], a popular reliable robustness evaluation benchmark. The robustness against other attacks is evaluated on the testing dataset against the same attack algorithms during fine-tuning.

### 5.2 Main Results

In this section, we present comprehensive experimental results to validate

**Superior Holistic Robustness.** On CIFAR-10, pretr_Linf model, CAS achieves the highest overall robustness (average = 52.54%), outperforming all baselines by at least 1.2 percentage points (vs. SAT + EMA 51.34%) while maintaining the highest clean precision (85.8%). Moreover, CAS establishes a new state-of-the-art for semantic robustness (52.52%), delivering at least 2.1% improvement over multi-attack baselines (E-AT-all+EMA: 50.42%).

**Cross-Model and Cross-Dataset Generalization.** While primarily tuning hyperparameters on CIFAR-10 with pretr_Linf models, we achieved competitive results using identical code on both pretr_L2, CIFAR-10 and pretr_Linf, CIFAR-100. Although individual metrics may not reach peak performance, our method consistently outperforms baselines in comprehensive evaluations. This demonstrates both the overall efficacy and transferability of our approach.

**MDE outperforms EMA in terms of multi-robustness.** Table 1 quantitatively demonstrates that integrating MDE with CAS yields a significant +0.69% improvement in average robustness (52.54% vs 51.85%) compared to CAS+EMA. This stems from MDE's organic integration of multidimensional characteristics and the dynamic nature of our CAS framework, which optimizes update weights for favorable training outcomes through dynamic decay rate selection and threshold-based filtering.

| Method | Clean | Avg. | avg. $\ell_p$ | avg. Sem. |
|---|---|---|---|---|
| pretr_Linf | 83.7 | 39.71 | 38.13 | 41.29 |
| E-AT | 83.1 | 50.66 | 54.70 | 46.62 |
| E-AT+EMA | 83.4 | 50.43 | 53.90 | 46.96 |
| E-AT-all | 84.1 | 51.13 | 52.07 | 50.18 |
| E-AT-all +EMA | 84.5 | 51.29 | 52.17 | 50.42 |
| RAMP | 84.2 | 50.68 | 54.93 | 46.43 |
| SAT | 85.0 | 50.83 | 51.87 | 49.79 |
| SAT+EMA | 84.1 | 51.34 | 53.93 | 48.74 |
| CAS | 85.0 | 51.81 | 51.90 | 51.72 |
| CAS+EMA | **85.9** | 51.85 | 51.9 | 51.81 |
| CAS+MDE | 85.8 | **52.54** | 52.57 | **52.52** |

Table 1: Overall camparison of our CAS+MDE method with baselines on CIFAR-10, pretr_Linf model.

| Method | Clean | Avg. | avg. $\ell_p$ | avg. Sem. |
|---|---|---|---|---|
| pretr_L2 | 88.2 | 39.43 | 41.53 | 37.32 |
| E-AT | 85.4 | 50.44 | 55.23 | 45.66 |
| E-AT +EMA | 84.9 | 50.99 | 55.50 | 46.49 |
| E-AT-all | 85.7 | 51.76 | 52.93 | 50.58 |
| E-AT-all +EMA | 86.6 | 52.16 | 53.17 | 51.15 |
| RAMP | 85.9 | 48.29 | 53.17 | 43.42 |
| SAT | 86.4 | 51.66 | 52.83 | 50.49 |
| SAT+EMA | 86.5 | 52.04 | 53.00 | 51.07 |
| CAS | **87.4** | 51.56 | 51.90 | 51.21 |
| CAS+MDE | 86.3 | **52.45** | 52.73 | **52.17** |

Table 2: Overall camparison of our CAS+MDE method with baselines on CIFAR-10, pretr_L2 model.

| Method | Clean | Avg. | avg. $\ell_p$ | avg. Sem. |
|---|---|---|---|---|
| pretr_Linf | 68.5 | 28.88 | 28.67 | 29.08 |
| E-AT | 67.0 | 29.92 | 33.67 | 26.17 |
| E-AT+EMA | 62.5 | 31.07 | 35.33 | 26.81 |
| E-AT-all | 69.0 | 31.33 | 31.83 | 30.83 |
| E-AT-all +EMA | 64.5 | 33.25 | 34.67 | 31.83 |
| SAT | 68.5 | 29.31 | 27.83 | 30.78 |
| SAT+EMA | 67.0 | 31.36 | 31.00 | 31.72 |
| CAS | 65.5 | 31.49 | 32.67 | 30.31 |
| CAS+MDE | 68.0 | 32.71 | 33.50 | 31.92 |

Table 3: Overall camparison of our CAS+MDE method with baselines on CIFAR-100, pretr_Linf model.

| Method | Clean | Avg. | avg. $\ell_p$ | avg. Sem. |
|---|---|---|---|---|
| AC | 84.7 | 51.51 | 52.27 | 50.75 |
| AC+MDE | 85.4 | 52.14 | 52.17 | 52.12 |
| AC+LO | 83.8 | 52.05 | 53.20 | 50.90 |
| AC+LO +MDE | 84.7 | 52.36 | 51.87 | 52.85 |
| AC+TR | 85.5 | 51.72 | 52.83 | 50.61 |
| AC+TR +MDE | 85.4 | 51.92 | 52.00 | 51.83 |
| AC+LO+TR (CAS) | 85.0 | 51.81 | 51.90 | 51.72 |
| CAS+MDE | 85.8 | 52.54 | 52.57 | 52.52 |

Table 4: Ablation study on calibrated probability weights (Eq. 9).

## 5.3 Ablation Study

In this subsection, we show the usefulness of each component of our CAS framework. All these experiments are conducted on pretrained-$\ell_\infty$ model, CIFAR-10.

**Loss- and Trade-off-Aware Weighting.** The superiority of our dynamic weighting over static sampling stems from its adaptive resource allocation capability. Our dynamic scheme actively redirects training focus toward attacks with:

- Low current robustness ($w_1(1 - acc[j])$ term)

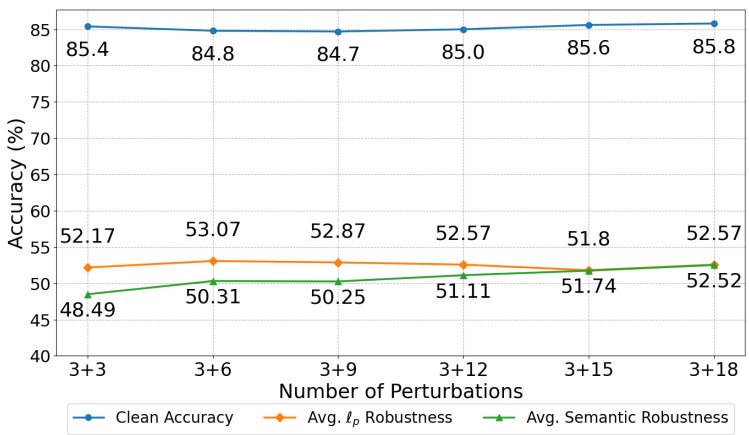

Figure 4: Ablation study on the number of considered perturbations.

- High learning difficulty ($w_2 loss[i][j]$ term)
- Positive systemic externalities ($w_3 V_{TR}$ term)

As evidenced in the Table 4, our holistic consideration of probability-weighted integration across all three factors yields more significant improvements compared to approaches using individual factors or pairwise combinations.

**Number of considered perturbations**. To investigate the stability and transferability of the CAS+MDE framework under varying perturbation regimes, we conduct fine-tuning experiments combining 3 $\ell_p$ attacks with progressively increasing numbers of semantic perturbations (3, 6, 9, 12, 15, 18). As shown in Figure 4, both clean accuracy and average $\ell_p$ robust accuracy remain remarkably stable across perturbation scales, while average semantic robust accuracy, which is still measured against all 18 semantic attacks post-fine-tuning, exhibits consistent improvement with additional perturbation types. This demonstrates exceptional method stability against configuration variations and confirms that incorporating more attack types can enhance overall robustness. Consequently, practical implementations would better maximize coverage of domain-relevant adversarial threats, while integrating and formalizing unforeseen attacks could further enrich multi-robustness benchmarks.

# 6 Conclusion

In this work, we provide a concise overview of multi-robustness research and its technical bottlenecks. We revisit the critical concept of Mutually Exclusive Perturbations (MEPs) and present an initial attempt to establish its quantitative definition. We holistically integrate and balance $\ell_p$ attacks and semantic attacks. Building on these foundations—along with dynamic probabilistic weighting and exponential moving average (EMA) variant—we propose Calibrated Adversarial Sampling (CAS), a fine-tuning framework that achieves state-of-the-art robustness across multiple adversarial settings. Extensive experiments validate the effectiveness of holistic multi-factor integration and the incorporation of diverse adversarial attack types within practical robust training pipelines. This establishes CAS as a strong baseline for multi-robustness fine-tuning and provides a promising paradigm for robust generalization.

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

# A The Complete Algorithm and Motivation of Our Method

---

**Algorithm 1:** CAS Framework

---

**Input:** Original DNN $N$, $epochs$,
batches of inputs and their targets $\{x_i, y_i\}_{i=1}^n$,
different adversarial attacks $\{A_1, A_2, \cdots, A_m\}$ and their epsilons $\epsilon_1, \epsilon_2, \cdots, \epsilon_m$

**Output:** A repaired DNN $N'$

**1** **Phase I: Sample-wise Testing**.
**2** **for** $j \leftarrow 1$ **to** $m$ **do**
**3**      evaluate the robust accuracy before training
**4**      **for** $i \leftarrow 1$ **to** $n$, **if** $i \equiv j \bmod m$ **do**
**5**          get the adversarial example $x'_{ij}$ of $x_i$ in $A_j$ attack
**6**          evaluate the loss[i][j] and accuracy of $x'_{ij}$
**7**      evaluate the robust accuracy before training
**8** get the total accuracy acc[j] for each attack $A_j$
**9** get the tradeoff matrix

**10** **Phase II: Warm Up**.
**11** **for** $epoch \leftarrow 1$ **to** $epochs$ **do**
**12**      **for** $i \leftarrow 1$ **to** $n$ **do**
**13**          **for** $j \leftarrow 1$ **to** $m$ **do**
**14**             compute the smoother probability weight based on acc[j](Eq. (5)).
**15**          balance the probability weights between $\ell_p$ attacks and semantic attacks(Eq. (6))
**16**          randomly choose attack $A_l$ for AT
**17**          update loss[i][j] and acc[j]
**18**      calculate the dynamic decay-rate and then deploy MDE average

**19** **Phase III: Dynamic Fine-tuning**.
**20** **for** $epoch \leftarrow 1$ **to** $epochs$ **do**
**21**      **for** $i \leftarrow 1$ **to** $n$ **do**
**22**          **for** $j \leftarrow 1$ **to** $m$ **do**
**23**             compute the calibrated probability weights(Eq. (9))
**24**          balance the probability weights and randomly choose $A_l$ for AT
**25**          update loss[i][j] and acc[j]
**26**      MDE with higher sensitivity

---

The complete algorithm of CAS+MDE method can be summarized in Algorithm 1. Overall, the CAS framework delivers qualitative superiority through its multi-phase design. First, sample-wise testing provides critical diagnostic precision by systematically evaluating robust accuracy and loss metrics for each attack type, thereby generating an innovative trade-off matrix that quantifies nuanced accuracy shifts between attack types. Transitioning to Phase II, the smooth probability weights prevent premature over-specialization by maintaining smoother attack-type distributions than steeper alternatives. The framework culminates in Phase III with its sophisticated multi-factor weighting system, which integrates three critical defense dimensions. This phase advances tradeoff modeling while implementing precision MDE with enhanced sensitivity ($d_3 > d_2$) to capture subtle late-stage improvements.

To investigate fine-tuning for multiple robustness more broadly, we augment the baseline that considers only $\ell_p$ attacks with semantic attacks. SAT's fixed probability weights are flawed, and E-AT's weights that focus solely on robust accuracy also fall short. We therefore adopt calibrated probability weights that integrate accuracy, loss, and trade-off. While studying the trade-offs in multiple robustness, we extend and quantify the notion of mutually exclusive perturbations. Recognizing that model-fusion techniques such as EMA are effective at boosting adversarial robustness and can be

plugged into many methods, we further design and incorporate MDE—an EMA variant tailored to the dynamic nature of our CAS approach.

## B    More Details about Our Experiments

At this point, we specify the hyperparameter selections for the main experiments with CIFAR-10 dataset:

| Parameter | Value |
|---|---|
| batch size | 128 |
| batch size for eval | 100 |
| max lr | 0.1 |
| weight decay | 5e-4 |
| decay rate (in EMA) | 0.995 |
| $w_1$ | 10 |
| $w_2$ | 1 |
| $w_3$ | 0.5 |
| $d_1$ | 5 |
| $d_2$ | 50 |
| $\tau$ (threshold in MDE) | 0.95 |

Table 5: Hyperparameters for CAS

## C    Additional Discussion on Experiment Results

By examining the detailed tabular data (Table 6, Table 7, and Table 8) from the "Overall Comparison" experiment in the main text, we can derive additional insights:

First, different types of adversarial attacks exhibit varying degrees of sensitivity to fine-tuning regarding robustness. For instance, the robust accuracy of L1 and prison attacks shows significant improvement across different fine-tuning approaches, whereas blur-based attacks demonstrate no noticeable changes in robust accuracy after fine-tuning.

Second, multi-robustness-based fine-tuning does not necessarily enhance robustness against all attack types. For example, the robust accuracy against hsv attacks decreases across all fine-tuning methods compared to the original model. This stems from complex interference patterns between different robustness types.

Third, the trends of improvement/decline and significance of changes in robust accuracy across all attack types remain largely consistent regardless of the fine-tuning method employed. This indicates that both the sensitivity of attack-specific robustness to fine-tuning and the trade-offs between different robustness types represent intrinsic properties, independent of fine-tuning approaches.

Fourth, we observe that robust accuracy against natural corruptions (*e.g.* fog, snow, blur) generally shows minimal variation after fine-tuning. In contrast, robustness against human-engineered adversarial attacks (*e.g.* $\ell_p$, JPEG, prison) exhibits relatively more pronounced changes post-fine-tuning.

Moreover, while we state in the main text that "To mitigate computational costs, Phase I executes just one epoch," detailed computational costs are not explicitly provided. In practice, our CAS method conducts one epoch for sample-wise testing, followed by 10 epochs of warm-up and 10 epochs of dynamic fine-tuning. All baselines undergo 20 epochs of fine-tuning. Under identical hardware configurations, the per-epoch computational time of our method shows minimal difference compared to probability-weighting-based baselines like E-AT and SAT. This demonstrates that our CAS method enhances efficacy without compromising efficiency.

## D    Cross-model Comparison

Meanwhile, we compared the performance of different pre-trained models under our multi-robustness metrics framework. The results (Table 9) reveal that the pretr_L1 model delivers the strongest overall

| Model | pretr_Linf | E-AT | E-AT +EMA | E-AT-all | E-AT-all +EMA | RAMP | SAT | SAT+EMA | CAS | CAS+EMA | CAS+MDE |
|---|---|---|---|---|---|---|---|---|---|---|---|
| Clean | 83.7 | 83.1 | 83.4 | 84.1 | 84.5 | 84.2 | 85 | 84.1 | 85 | 85.9 | 85.8 |
| Avg. | 39.71 | 50.66 | 50.43 | 51.13 | 51.29 | 50.68 | 50.83 | 51.34 | 51.81 | 51.85 | 52.54 |
| avg. $\ell_p$ | 38.13 | 54.70 | 53.90 | 52.07 | 52.17 | 54.93 | 51.87 | 53.93 | 51.90 | 51.90 | 52.57 |
| avg. Sem. | 41.29 | 46.62 | 46.96 | 50.18 | 50.42 | 46.43 | 49.79 | 48.74 | 51.72 | 51.81 | 52.52 |
| Linf | 48.1 | 43.2 | 41.7 | 39.3 | 40.2 | 46.8 | 38.5 | 41.3 | 40.3 | 39.8 | 41 |
| L2 | 60 | 67 | 67 | 67.2 | 67.5 | 67.9 | 67.5 | 67.6 | 67.3 | 67.7 | 68.1 |
| L1 | 6.3 | 53.9 | 53 | 49.7 | 48.8 | 50.1 | 49.6 | 52.9 | 48.1 | 48.2 | 48.6 |
| ppgd | 42.5 | 53.9 | 53.5 | 50.7 | 50.6 | 55.3 | 50.8 | 53 | 51.2 | 50.9 | 52.2 |
| wood | 58.8 | 72 | 71.7 | 73.9 | 72.6 | 71.8 | 71.9 | 72.6 | 75.7 | 74.2 | 74.7 |
| elastic | 28.3 | 29.8 | 28.4 | 37.4 | 38.9 | 29.3 | 38.2 | 32.3 | 37.8 | 38 | 39 |
| pixel | 27.8 | 38.8 | 43.7 | 38.8 | 41.1 | 35.5 | 41.4 | 38.2 | 37.9 | 40.1 | 39.2 |
| snow | 24.9 | 21.5 | 20.7 | 21.9 | 24.8 | 18.9 | 26.8 | 22 | 26.1 | 24.3 | 26.6 |
| gabor | 38.6 | 45.9 | 49.9 | 50.6 | 39.6 | 47.4 | 38 | 51.3 | 65.6 | 56 | 58.8 |
| jpeg | 44.6 | 59.5 | 60.8 | 58.4 | 58.1 | 60 | 57.2 | 60.3 | 58.8 | 58.7 | 59.3 |
| glitch | 30.2 | 38.8 | 36.5 | 39.5 | 40.7 | 36.3 | 41.4 | 38.6 | 40.6 | 40.2 | 41 |
| kalei | 39.2 | 44.1 | 42.1 | 56.3 | 59.9 | 40.7 | 58.9 | 50.1 | 56.6 | 58.4 | 58.8 |
| blur | 54.2 | 52 | 52 | 55.5 | 56 | 52.6 | 56.7 | 53.9 | 53.7 | 57.3 | 56.1 |
| edge | 44.2 | 52.9 | 54.1 | 58.7 | 57.9 | 55 | 57.1 | 56.1 | 59.9 | 59.3 | 59.6 |
| fog | 73.4 | 69.3 | 70.9 | 71.9 | 73.2 | 71.9 | 72.5 | 72.9 | 73.3 | 75.4 | 74.5 |
| texture | 34.1 | 42.2 | 40.5 | 45.8 | 47.2 | 41 | 46.6 | 44 | 48.9 | 45.2 | 47.7 |
| prison | 31.8 | 52.4 | 50.8 | 58.5 | 57.5 | 49.2 | 50.5 | 51.9 | 57.6 | 59.8 | 62 |
| whirl | 55.3 | 58.9 | 62.1 | 67.7 | 67.2 | 59.4 | 66.5 | 66.8 | 67.7 | 69.2 | 67.4 |
| polkadot | 33.5 | 28.5 | 30.6 | 32.8 | 34.3 | 30.3 | 33 | 33.1 | 31.8 | 36.6 | 37 |
| klotski | 30.5 | 33.9 | 33.5 | 41.5 | 42.8 | 32.5 | 43.4 | 36.1 | 43.5 | 44.4 | 45.1 |
| hsv | 51.4 | 44.8 | 43.5 | 43.4 | 45.1 | 48.6 | 45.3 | 44.2 | 44.3 | 44.5 | 46.4 |

Table 6: Overall camparison of our CAS+MDE method with baselines on CIFAR-10, pretr_Linf model.

| Model | pretr_L2 | E-AT | E-AT +EMA | E-AT-all | E-AT-all +EMA | RAMP | SAT | SAT+EMA | CAS | CAS+EMA | CAS+MDE |
|---|---|---|---|---|---|---|---|---|---|---|---|
| Clean | 88.2 | 85.4 | 84.9 | 85.7 | 86.6 | 85.9 | 86.4 | 86.5 | 87.4 | 87.4 | 86.3 |
| Avg. | 39.43 | 50.44 | 50.99 | 51.76 | 52.16 | 48.29 | 51.66 | 52.04 | 51.56 | 52.02 | 52.45 |
| avg. $\ell_p$ | 41.53 | 55.23 | 55.50 | 52.93 | 53.17 | 53.17 | 52.83 | 53.00 | 51.90 | 51.87 | 52.73 |
| avg. Sem. | 37.32 | 45.66 | 46.49 | 50.58 | 51.15 | 43.42 | 50.49 | 51.07 | 51.21 | 52.18 | 52.17 |
| Linf | 29.8 | 41.5 | 41.6 | 39 | 38.8 | 43.6 | 38.2 | 38.1 | 38.5 | 37.4 | 38.5 |
| L2 | 68.6 | 70 | 70.1 | 69.8 | 70 | 69.6 | 69.5 | 70.1 | 69.9 | 70.1 | 71.4 |
| L1 | 26.2 | 54.2 | 54.8 | 50 | 50.7 | 46.3 | 50.8 | 50.8 | 47.3 | 48.1 | 48.3 |
| ppgd | 43.7 | 54.8 | 54.9 | 52.1 | 53.2 | 54.8 | 52.4 | 52.6 | 51.4 | 52.4 | 52.3 |
| wood | 70 | 71.5 | 73.1 | 71.2 | 75.7 | 73.9 | 75.8 | 76.7 | 77.5 | 76.5 | 75.6 |
| elastic | 14 | 28.7 | 28.4 | 38.7 | 38.4 | 25.2 | 36.8 | 35.9 | 39 | 37.1 | 39.2 |
| pixel | 21.7 | 43.8 | 43.8 | 51.4 | 40.9 | 31.8 | 40.6 | 40.7 | 38.5 | 46.7 | 41 |
| snow | 9.7 | 17.4 | 18.7 | 24.9 | 27 | 14.7 | 26.3 | 27.8 | 22.4 | 23.2 | 22.8 |
| gabor | 32.6 | 23.5 | 41.8 | 34 | 43.7 | 29 | 39 | 40.9 | 43.1 | 50.4 | 51.9 |
| jpeg | 52.2 | 61.3 | 61.5 | 58.5 | 59.7 | 60.9 | 60.4 | 60.1 | 60.9 | 60.2 | 59.9 |
| glitch | 20.9 | 38.2 | 37.3 | 39.1 | 40.5 | 32.2 | 40.3 | 39.8 | 38.8 | 40.3 | 40.6 |
| kalei | 34.1 | 41.4 | 40 | 61.4 | 60.2 | 39.2 | 61.2 | 60.1 | 62.7 | 62.7 | 63.4 |
| blur | 55.4 | 53.9 | 52.5 | 57.5 | 55.9 | 52.7 | 57.6 | 58.3 | 56.9 | 59.4 | 60.9 |
| edge | 42.1 | 53.8 | 53.8 | 57.3 | 59 | 51.8 | 58.1 | 58.4 | 58.2 | 59.7 | 58.9 |
| fog | 77.8 | 72.3 | 71.5 | 73.9 | 74 | 73 | 75.2 | 74.9 | 73.4 | 76.1 | 76.2 |
| texture | 21.5 | 38.8 | 38.6 | 44.6 | 43.8 | 38.2 | 44.7 | 43.1 | 44.9 | 42.2 | 42.5 |
| prison | 26.8 | 51.5 | 51.1 | 55.6 | 55.5 | 42.1 | 51.5 | 56.9 | 58.8 | 57.5 | 57.6 |
| whirl | 54.3 | 62.5 | 61.7 | 68 | 70.2 | 58.8 | 68.6 | 69.4 | 70.3 | 69 | 68.5 |
| polkadot | 40.6 | 32.6 | 31.7 | 33.2 | 34 | 31 | 34 | 36.1 | 35.5 | 36.9 | 36.7 |
| klotski | 19.1 | 31.8 | 31.7 | 43.4 | 44.1 | 28.5 | 41.7 | 42 | 45 | 43.7 | 45.8 |
| hsv | 35.3 | 44 | 44.7 | 45.7 | 44.9 | 43.8 | 44.7 | 45.6 | 44.5 | 45.2 | 45.2 |

Table 7: Overall camparison of our CAS+MDE method with baselines on CIFAR-10, pretr_L2 model.

performance. The pretr_L2 model exhibits deficiencies in robustness against semantic attacks and shows significant weaknesses when facing adversarial attacks like elastic and snow. The pretr_Linf model underperforms in both clean accuracy and $\ell_p$ robustness, primarily due to the pronounced trade-offs between $\ell_\infty$ robustness versus clean accuracy and $\ell_1$ robustness.

This suggests that if adversarial training focuses solely on a single perturbation type while aiming for favorable clean accuracy and multi-robustness metrics, $\ell_1$-based adversarial attacks may be a preferable choice. However, in practice, the $\ell_\infty$ model often holds certain advantages. For example, $\ell_\infty$ pre-trained models generally demonstrate superior defense capability against highly engineered adversarial attacks (*e.g.* hsv and klotski) ; and $\ell_\infty$ adversarial training methods (*e.g.* PGD-AT for $\ell_\infty$) are simpler and more mature than their $\ell_1$ counterparts (e.g., APGD-AT for $\ell_1$).

| Model | pretr_Linf | E-AT | E-AT +EMA | E-AT-all | E-AT-all +EMA | SAT | SAT+EMA | CAS | CAS+MDE |
|---|---|---|---|---|---|---|---|---|---|
| **Clean** | 68.5 | 67 | 62.5 | 69 | 64.5 | 68.5 | 67 | 65.5 | 68 |
| **Avg.** | 28.88 | 29.92 | 31.07 | 31.33 | 33.25 | 29.31 | 31.36 | 31.49 | 32.71 |
| **avg. $\ell_p$** | 28.67 | 33.67 | 35.33 | 31.83 | 34.67 | 27.83 | 31.00 | 32.67 | 33.50 |
| **avg. Sem.** | 29.08 | 26.17 | 26.81 | 30.83 | 31.83 | 30.78 | 31.72 | 30.31 | 31.92 |
| **Linf** | 31.5 | 23 | 24.5 | 21 | 25 | 17.5 | 21 | 23.5 | 22 |
| **L2** | 40 | 45 | 46 | 46 | 46.5 | 44 | 44.5 | 46 | 48 |
| **L1** | 14.5 | 33 | 35.5 | 28.5 | 32.5 | 22 | 27.5 | 28.5 | 30.5 |
| **ppgd** | 36 | 34.5 | 35.5 | 31.5 | 33.5 | 28.5 | 32.5 | 31.5 | 33 |
| **wood** | 42.5 | 50.5 | 51.5 | 54.5 | 53 | 49 | 53.5 | 51.5 | 57 |
| **elastic** | 24.5 | 14.5 | 20.5 | 21.5 | 22 | 23.5 | 23.5 | 22 | 23.5 |
| **pixel** | 21.5 | 19 | 20 | 23 | 23 | 24.5 | 24.5 | 25.5 | 24.5 |
| **snow** | 19 | 9 | 10.5 | 10 | 14.5 | 10.5 | 17.5 | 12 | 11.5 |
| **gabor** | 7.5 | 7.5 | 17 | 20.5 | 34.5 | 13.5 | 16.5 | 14.5 | 16 |
| **jpeg** | 34 | 40 | 41.5 | 39.5 | 40.5 | 33.5 | 38 | 38.5 | 39 |
| **glitch** | 20 | 17.5 | 17 | 20.5 | 20 | 21 | 19 | 20.5 | 21.5 |
| **kalei** | 25.5 | 21 | 21.5 | 42.5 | 39.5 | 41.5 | 45 | 38.5 | 40.5 |
| **blur** | 40.5 | 35.5 | 31.5 | 35 | 33 | 43 | 35 | 42.5 | 36 |
| **edge** | 31 | 32.5 | 35.5 | 37.5 | 45.5 | 35 | 45 | 35.5 | 41 |
| **fog** | 58 | 53 | 46 | 55.5 | 48 | 60 | 52.5 | 55 | 54.5 |
| **texture** | 24 | 15 | 17 | 17 | 26.5 | 14 | 27.5 | 19 | 25.5 |
| **prison** | 15 | 21 | 17.5 | 28.5 | 26 | 21.5 | 23 | 21 | 24.5 |
| **whirl** | 42.5 | 42 | 38 | 46 | 46 | 51 | 45.5 | 47 | 52.5 |
| **polkadot** | 20.5 | 17 | 16.5 | 22 | 14.5 | 29.5 | 16.5 | 22 | 18 |
| **klotski** | 24.5 | 18.5 | 19 | 25 | 24.5 | 30 | 29 | 24.5 | 30 |
| **hsv** | 37 | 23 | 26.5 | 25 | 28.5 | 24.5 | 27 | 24.5 | 26 |

Table 8: Overall camparison of our CAS+MDE method with baselines on CIFAR-100, pretr_Linf model.

# E    Extended Experiments on Multi-Robustness Trade-offs

## E.1    Semantic-$\ell_p$ Conflicts

As quantified in the tradeoff matrix (Figure 5), semantic attacks generally degrade $\ell_p$ robustness due to incompatible perturbation constraints, but exhibit texture-dependent exceptions. Weather corruptions (*e.g.* fog and snow) conflict universally with all $\ell_p$ norms.

Moreover, we find that Linf robustness suffers much more overall semantic interference than L1 robustness due to fundamental differences in perturbation geometry.

## E.2    Transfer Asymmetry

While $\ell_p$ training before semantic attacks yields net positive transfer, the reverse sequence causes disproportionate degradation. Strikingly, elastic $\rightarrow$ wood transfer (+0.192) demonstrates notably greater efficacy than the reverse direction (-0.007), indicating a broad and pronounced asymmetry in the trade-offs between different types of robustness.

## E.3    Robustness Interference

The significantly negative matrix sum ($\sum_{i,j} M_{ij} = -3.123$) confirms global interference, but masks polarized cluster dynamics. We also computed the row-wise and column-wise sums of the trade-off matrix. Under single-type training, we observed that—apart from pixel-level perturbations—every semantic attack in the table reduces the overall robust-accuracy sum; adversarial training against the gabor attack even degrades robustness to all other attack types. In contrast, adversarial training on the three $\ell_p$ norms consistently improves overall robustness, underscoring why practical DNN robust-training regimes usually focus on $\ell_P$ attacks. Only wood and gabor robust-accuracy achieve noticeable overall gains when adversarial training is progressively applied to each perturbation

| Model | pretr_Linf | pretr_L2 | pretr_L1 |
|---|---|---|---|
| **Clean** | 83.7 | 88.2 | 87.1 |
| **Avg.** | 39.71 | 39.43 | 45.30 |
| **avg. $\ell_p$** | 38.13 | 41.5 | 48.97 |
| **avg. Sem.** | 41.29 | 37.32 | 41.63 |
| **Linf** | 48.1 | 29.8 | 22 |
| **L2** | 60 | 68.6 | 64.9 |
| **L1** | 6.3 | 26.2 | 60 |
| **ppgd** | 42.5 | 43.7 | 43.3 |
| **wood** | 58.8 | 70 | 73.6 |
| **elastic** | 28.3 | 14 | 14.8 |
| **pixel** | 27.8 | 21.7 | 30.2 |
| **snow** | 24.9 | 9.7 | 14.1 |
| **gabor** | 38.6 | 32.6 | 41.6 |
| **jpeg** | 44.6 | 52.2 | 47.1 |
| **glitch** | 30.2 | 20.9 | 30.5 |
| **kalei** | 39.2 | 34.1 | 35 |
| **blur** | 54.2 | 55.4 | 56 |
| **edge** | 44.2 | 42.1 | 53.3 |
| **fog** | 73.4 | 77.8 | 75.7 |
| **texture** | 34.1 | 21.5 | 24.3 |
| **prison** | 31.8 | 26.8 | 54.4 |
| **whirl** | 55.3 | 54.3 | 62.4 |
| **polkadot** | 33.5 | 40.6 | 44.7 |
| **klotski** | 30.5 | 19.1 | 20.5 |
| **hsv** | 51.4 | 35.3 | 27.9 |

Table 9: Camparison of 3 pretrained PreActResNet-18 models based on our multi-robustness benchmark

type, highlighting the formidable challenge that robustness inference poses to improving multiple robustness dimensions simultaneously.

This explains CAS's effectiveness: stochastic sampling prevents irreversible catastrophic damage from "interference source" attacks (gabor $\rightarrow$ fog : $-0.384$) by distributing training across compatibility zones.

# F  Ablation Studies

## F.1  Number of Considered Perturbations

In our ablation study on the number of considered perturbations, we gradually add new semantic attacks to the fine-tuning pipeline from top to bottom, adding three at each step. The detailed results show that most robustness types do not improve significantly when their corresponding attacks are included; only gabor and Kaleidoscope enjoy noticeable robust-accuracy gains. Moreover, introducing new attacks can reduce the robustness of previously considered ones—for example, pixel robust-accuracy drops sharply after the final three attacks are added. Overall, however, considering a wider range of perturbations benefits overall multiple robustness. We also observe that clean accuracy and $\ell_p$ robustness remain relatively stable as the attack count grows, because we balance the probability weights of $\ell_p$ and semantic attacks during fine-tuning.

$\ell_p$ **and semantic reweighting**. Table 10 demonstrates that balancing our probability weights between $\ell_p$ and semantic attacks significantly enhances average robustness. The motivation for increasing the relative weight of $\ell_p$ attacks stems from empirical observations in E-AT and RAMP: adversarial training focused solely on $\ell_p$ threats unexpectedly improves semantic robustness [6, 14]—though substantially less than direct semantic AT. Our trade-off matrix quantitatively validates this phenomenon, revealing that $\ell_p$ adversarial training enhances most semantic robustness while semantic

| | Linf | L2 | L1 | wood | elastic | pixel | snow | gabor | jpeg | blur | fog | sum |
|---|---|---|---|---|---|---|---|---|---|---|---|---|
| Linf | 0.002 | 0 | -0.009 | 0.017 | 0.003 | 0.001 | -0.001 | 0 | -0.018 | 0.014 | 0.005 | 0.014 |
| L2 | -0.001 | 0.006 | -0.005 | 0.004 | -0.007 | -0.028 | 0.010 | 0.046 | -0.006 | 0.005 | 0.004 | 0.028 |
| L1 | 0.010 | 0.008 | 0.026 | 0.033 | -0.005 | -0.025 | 0 | 0.074 | 0.019 | -0.035 | -0.031 | 0.074 |
| wood | -0.009 | 0.006 | -0.002 | 0.014 | -0.007 | 0.013 | 0.006 | -0.090 | -0.017 | 0.006 | 0.009 | -0.071 |
| elastic | -0.087 | -0.057 | -0.036 | 0.192 | 0.082 | -0.102 | -0.063 | 0.228 | -0.033 | -0.159 | -0.060 | -0.095 |
| pixel | -0.013 | -0.015 | -0.008 | 0.013 | -0.006 | 0.039 | -0.001 | 0.081 | -0.034 | -0.023 | -0.003 | 0.030 |
| snow | -0.036 | -0.032 | -0.029 | -0.030 | -0.047 | -0.010 | 0.035 | 0.048 | -0.050 | 0.038 | 0.025 | -0.088 |
| gabor | -0.250 | -0.258 | -0.097 | -0.118 | -0.156 | -0.302 | -0.008 | 0.075 | -0.124 | -0.334 | -0.384 | -1.956 |
| jpeg | -0.072 | -0.058 | -0.040 | 0.147 | -0.101 | -0.114 | -0.052 | 0.097 | 0.029 | -0.249 | -0.133 | -0.546 |
| blur | -0.008 | 0.007 | -0.002 | 0.061 | -0.017 | -0.019 | 0.003 | -0.073 | 0.004 | 0.013 | 0 | -0.031 |
| fog | -0.104 | -0.096 | -0.098 | -0.026 | -0.068 | 0.044 | -0.066 | -0.068 | -0.141 | 0.090 | 0.051 | -0.482 |
| sum | -0.568 | -0.489 | -0.300 | 0.307 | -0.329 | -0.503 | -0.137 | 0.418 | -0.371 | -0.634 | -0.517 | |

Figure 5: Trade-off matrix. In the figure, each number represents the change in robust accuracy for its corresponding attack type after AT against designated types. Each row shows the data fine-tuned against the adversarial attack labeled on the left; each column shows the data evaluated after being tested with the attack labeled above.

adversarial training often impairs $\ell_p$ robustness. This asymmetric relationship fundamentally informs our reweighting strategy.

| Method | Clean | Avg. | avg. $\ell_p$ | avg. Sem. |
|---|---|---|---|---|
| no reweight | 85.6 | 48.84 | 46.27 | 56.42 |
| no reweight+MDE | 86 | 50.44 | 47.47 | 53.42 |
| CAS | 85.0 | 51.81 | 51.90 | 51.72 |
| CAS+EMA | 85.8 | **52.54** | **52.57** | 52.52 |

Table 10: Ablation study on reweighting (Eq. 6).

## F.2 Components in Calibrated Probability Weights

Regarding the Ablation Study results on Calibrated Probability Weights, where CAS(AC+LO+TR) exhibits lower average accuracy than AC+LO+MDE, AC+TR+MDE, and AC+LO, we clarify that this does not imply mutual exclusivity between LO and TR. The apparent discrepancy arises because MDE inherently elevates clean and robust accuracy within adversarial robustness frameworks. Consequently, comparing the plain CAS configuration (AC+LO+TR) against variants augmented with MDE (e.g., AC+LO+MDE) constitutes an unbalanced comparison. Crucially, LO and TR operate synergistically in the probability weighting mechanism: while LO alone typically increases robust accuracy at the expense of clean accuracy, the combined integration of LO and TR enables more effective balancing of these objectives. This synergy ultimately enhances both clean and robust accuracies, demonstrating the significance of the full AC+LO+TR integration in our proposed CAS framework.

# G Limitations

While our trade-off matrix provides a quantitative definition of Mutually Exclusive Perturbations (MEPs), this framework and related investigations remain insufficiently rigorous. Critical questions persist: (i) Model selection: On which model architectures should MEPs be evaluated? (ii) Perturbation calibration: How should perturbation intensities be standardized across diverse adversarial attacks?

|          | 3+3   | 3+6   | 3+9   | 3+12  | 3+15  | 3+18  |
|----------|-------|-------|-------|-------|-------|-------|
| **Clean** | 85.4  | 84.8  | 84.7  | 85    | 85.6  | 85.8  |
| **Avg.**  | 50.33 | 51.69 | 51.56 | 51.84 | 51.77 | 52.54 |
| **avg. $\ell_p$** | 52.17 | 53.07 | 52.87 | 52.57 | 51.80 | 52.57 |
| **avg. Sem.** | 48.49 | 50.31 | 50.25 | 51.11 | 51.74 | 52.52 |
| **Linf** | 40.1  | 41.3  | 41.6  | 40.5  | 39.1  | 41    |
| **L2**   | 68    | 68.5  | 67.9  | 67.8  | 67.4  | 68.1  |
| **L1**   | 48.4  | 49.4  | 49.1  | 49.4  | 48.9  | 48.6  |
| **ppgd** | 52.3  | 54    | 52.3  | 52.3  | 52    | 52.2  |
| **wood** | 73.6  | 72.4  | 73.1  | 72.2  | 74.3  | 74.7  |
| **elastic** | 40.3 | 41.5 | 38.3 | 37.9 | 35.8 | 39 |
| **pixel** | 43.3 | 44.5 | 50.6 | 46.7 | 46.6 | 39.2 |
| **snow** | 26.3 | 25.4 | 25.1 | 27.5 | 23.2 | 26.6 |
| **gabor** | 38.7 | 65.9 | 40.3 | 56.3 | 54.1 | 58.8 |
| **jpeg** | 58.7 | 58.3 | 59.9 | 59.7 | 59 | 59.3 |
| **glitch** | 37.8 | 39.5 | 40.6 | 42.4 | 41.5 | 41 |
| **kalei** | 42 | 43.2 | 63.9 | 62.2 | 62.7 | 58.8 |
| **blur** | 54.1 | 55.6 | 54.3 | 55.2 | 57.8 | 56.1 |
| **edge** | 57.2 | 55.6 | 55.8 | 57.9 | 58.3 | 59.6 |
| **fog** | 72.5 | 72.6 | 72.6 | 72.9 | 74.1 | 74.5 |
| **texture** | 40.2 | 40.6 | 42.4 | 41.5 | 44.4 | 47.7 |
| **prison** | 48.6 | 47.3 | 49.8 | 52.8 | 56.9 | 62 |
| **whirl** | 66.3 | 65.8 | 68.1 | 64.1 | 67.9 | 67.4 |
| **polkadot** | 33.8 | 33.6 | 31.9 | 32.4 | 39.2 | 37 |
| **klotski** | 43 | 44.2 | 41.6 | 41.9 | 40.7 | 45.1 |
| **hsv** | 44.1 | 45.5 | 43.9 | 44.1 | 42.8 | 46.4 |

Table 11: Ablation study on the number of considered perturbations.

Furthermore, translating theoretical trade-off insights into direct robustness enhancement remains an underexplored yet promising research direction. Our ablation study confirms that incorporating trade-off relationships into probabilistic weighting yields measurable benefits, though its efficacy is less pronounced than accuracy-based adaptations.

Notably, while CAS achieves state-of-the-art results in both clean accuracy and average robust accuracy, our MEP analysis reveals a fundamental tension: enhancing robustness against mutually exclusive perturbations inherently incurs costs in clean accuracy or other robustness dimensions. This suggests potential degradation against unconsidered attack types—an intrinsic challenge in multi-robustness fine-tuning. To mitigate this limitation, we could systematically identify and cover those attack types prevalent in specific application areas before deploying such frameworks in real-world scenarios.

## H   Promising Research Directions

Finally, we reflect on the study's limitations and outline promising research directions:

1. Advancing trade-off characterization: Rigorous benchmarking frameworks or novel theoretical frameworks for multi-robustness trade-offs are waiting to be developed.

2. Expanding attack typology: New adversarial constraints could be proposed, or existing semantic attacks refined through granular categorization to clarify robustness conflicts/synergies.

3. Exploring composite perturbations: Stronger attacks may emerge from strategically combined perturbation types.

4. Formalizing non-algorithmic attacks: Attacks lacking clear algorithmic definitions warrant alternative formalizations or methodological innovations for systematic study.

