# OpenReview forum: "Generalizing Robustness from $\ell_p$ to Unforeseen Attack via Calibrated Adversarial Sampling"
_NeurIPS.cc/2025/Workshop/Reliable_ML — NeurIPS 2025 - Reliable ML Workshop_

### Official Review · Reviewer_bgeC · 2025-09-14
**Good experimental performance, lack of clarity at certain parts**

**Rating:** 6
**Confidence:** 3

**Review:**

-- Summary. This paper proposes a new fine-tuning method that enhances the robustness of DNNs to adversarial perturbations of multiple types. In particular, the authors consider both $\ell_p$ perturbations on the input space, as well as semantic perturbations, like, for example, rotations of images. While there are adversarial training methods that achieve strong robustness to multiple types of perturbations, they require more computational resources than fine-tuning methods. The authors evaluate the performance of their method experimentally, showcasing state-of-the-art performance on robustness to semantic perturbations, as well as overall robustness averaged over both $\ell_p$ and semantic perturbations.

-- Strengths. The problem of adversarial robustness is important and this paper proposes a practical fine-tuning method with favorable performance. Moreover, the authors explain why achieving multi-robustness is a challenging task, by performing a quantitative analysis of the multi-robustness trade-offs.

-- Weaknesses. Even though the concepts and algorithms discussed in this paper are not too complicated, I found that there is a lack of clarity in some parts:
1. For example, in section 4.2 the authors use notation that has not been defined previously (e.g. $x_i, D_j$).
2. Moreover, the authors mention robust accuracy "before" and "after" training, but it is not clear what "training" refers to. While Algorithm 1 in the appendix does introduce the required notation ($x_i$ is an input feature vector), it remains unclear what "robust accuracy before/after training" refers to.
3. In lines 9, 18, 26 of Algorithm 1 the authors refer to notions defined in the main part ("tradeoff matrix", "MDE"). It would be clearer if they defined these notions in place, so that the reader does not need to move back and forth. At the very least, the authors could use the Definition environment to define these notions elsewhere and refer to the appropriate definition whenever such a notion is mentioned in Algorithm 1.

Moreover, another potential weakness is that the algorithm targets a fixed and finite (albeit diverse) set of candidate perturbations. In particular, in lines 261-262, the authors mention that the semantic robustness is evaluated on the same types of semantic attacks that were used for fine-tuning. It would be interesting to explore if fine-tuning with respect to one type of semantic attack implies some level of robustness with respect to some other type as well.

---

### Official Review · Reviewer_kShN · 2025-09-20
**Comment**

**Rating:** 6
**Confidence:** 4

**Review:**

The paper introduces Calibrated Adversarial Sampling to improve the robustness of DNNs against unforeseen attacks by dynamically adjusting sampling probabilities.


Pros: The experiments are well set, and the method is easy to interpret and apply.

Cons: The paper does not give a concrete mathematical definition and analysis. Experiments are small scale.